# ROLoRA: Rank Optimization for Low-Rank Adaptation under Memory Constraints

## Abstract

Low-Rank Adaptation (LoRA) has emerged as a prominent technique for fine-tuning large language models (LLMs) with limited computational resources. However, by injecting low-rank adapters with a rank identical across all layers, standard LoRA overlooks the varying importance of the weight matrices, often leading to suboptimal performance. Therefore, discovering an optimal rank configuration that efficiently utilizes limited training resources remains an open question. Existing solutions typically compromises computational constraints for performance gains, limiting their practical usage in resource-constrained scenarios. To address these issues, in this paper, we propose a novel method named ROLoRA to efficiently discover an effective rank configuration for low-rank adaptation, while strictly adhering to a constrained computational budget during training. In particular, our method iteratively prunes saturated adapters and expands underfitted ones to increase their capacity until they converge to a highly optimized configuration. Our approach is delicately designed within the Frank-Wolfe algorithmic framework, which offers potential theoretical guarantees. Experimentally, we demonstrate that ROLoRA outperforms standard LoRA on common natural language processing tasks, including the GLUE and SQuAD benchmarks. Additionally, we provide a comprehensive analysis to explain why ROLoRA surpasses competing state-of-the-arts.

## 1 Introduction

Fine-tuning pre-trained large language models (LLMs) for downstream tasks has become a common practice to meet customized and domain-specific demands, especially after the recent prevalence of these models. However, the computational cost during training remains a significant concern, particularly due to the increasing number of parameters in these models. Naively fine-tuning the entire models often requires as much computational capacity as training the model from scratch, which limits the feasibility of fine-tuning LLMs on resource-constrained edge devices.

To fully exploit the advantages of model fine-tuning, recent works have departed from full fine-tuning towards more light-weighted approaches, also known as Parameter-Efficient Fine-Tuning methods (or PEFT) (Houlsby et al., 2019; Karimi Mahabadi et al., 2021; Mao et al., 2022). PEFT methods aim at reducing the number of trainable parameters during the fine-tuning process, which brings about simultaneously two benefits. First, PEFT methods reduce the memory required to store trained parameters, enabling the deployment of fine-tuned models for multiple tasks on the same device. Second, the reduced number of trainable parameters typically lowers the optimization cost during training, particularly by minimizing the memory required to store optimizer's states, which facilitates on-device fine-tuning for these models.

Among many recently developed PEFT methods, Low-Rank Adaptation, or LoRA, proposed by Hu et al. (2022) is among the most important. In principle, LoRA freezes the base pre-trained model and attach updates to each weight matrix of the base model as separate modules, known as adapters, in form of a product of two matrices of significantly smaller size. Concretely,

$$W = W_0 + \Delta W = W_0 + BA,$$

where $W_0 \in \mathbb{R}^{n \times d}$ is the base pre-trained weight, $\Delta W = BA$ is the update, $A \in \mathbb{R}^{r \times d}$ is called the down projection matrix and $B \in \mathbb{R}^{n \times r}$ the up projection matrix. Typically the size $r$ is chosen

to be a small constant significantly smaller than $n, d$. In this way, LoRA is able to reduce the number of trainable parameters to possibly only 0.5% compared with full fine-tuning while achieves comparable or even better than the latter (Hu et al., 2022; Zhang et al., 2023).

As simple and efficient as it may seem, LoRA still has its limitations. In particular, it is unclear how to optimally choose the rank $r$ for each adapter in LoRA. The standard LoRA sets an identical rank $r$ for all adapters. This strategy ignores the fact that different layers in the model carry different amount of knowledge (Chen et al., 2023a) and thus require different level of tuning. In a recent work, Zhang et al. (2023) demonstrate that varying the ranks $r$ according to the importance of the weight matrices can improve the performance of LoRA-style methods. However, existing solutions often require a greater memory budget during training to provide more buffer to explore a high-performing adapter configuration. This is especially undesirable for fine-tuning the model on small devices where the memory allowed during training is strictly limited.

In this work, we study the problem of determining the ranks of the adapters that can improve the performance of LoRA while imposing a strict memory constraint during training, reflected by a constraint on the number of trainable parameters. Towards answering this question, our contributions can be summarized as follows.

We propose a general framework, named ROLoRA, that iteratively sparsifies and grows the ranks of the adapters. Specifically, at each iteration, our algorithm performs a flexible rank sparsification step to determine the importance of the adapted weight matrices. Our framework allows to use any rank sparsification algorithm as a blackbox. Next, we expand the rank configuration to increase the capacity of the adapters, while strictly adhering to memory budget, before updating the ranks following the direction suggested this configuration. Our approach aims to imitate a first order gradient method (specifically Frank-Wolfe algorithm), which is search-free and highly efficient.

To demonstrate its effectiveness, we show in experiment that our algorithm improves the performance of LoRA on common NLP benchmarks, including GLUE (Wang et al., 2019) and SQUAD datasets (Rajpurkar et al., 2016; 2018). We show that our algorithm even outperforms AdaLoRA which requires more memory during training. We show at the same time that the output adapters can be smaller in size, reducing the memory storage compared with the standard LoRA.

Finally, to explain the effectiveness of our method, we conduct a comprehensive ablation study. We demonstrate that a certain type of weight matrices in the transformer architecture, such as value matrices, plays a more important role than the others (key and query matrices) in fine-tuning. Meanwhile, our approaches could effectively discover the varying saliency pattern and assign proper ranks accordingly. These advantages contribute to the superior performance we achieved.

## 2 RELATED WORK

**Parameter-Efficient Fine-Tuning**. Together with model compression (Jafari et al., 2021; Chen et al., 2021), PEFT is another category of solutions to improving fine-tuning LLMs. One approach to PEFT is adding adapters between model layers (Houlsby et al., 2019; Rebuffi et al., 2017; Pfeiffer et al., 2021; He et al., 2022). Due to the increase in the number of layers, this approach, however, can introduce latency during inference time. Another direction to PEFT is to directly update the pre-trained weights, under which falls LoRA (Hu et al., 2022). We refer the reader to a recent survey on PEFT methods by Han et al. (2024) for a more comprehensive comparison.

**LoRA and related works**. We provide backgrounds on LoRA in Section 3. This seminal work by Hu et al. (2022) opens up a plethora of works on PEFT via low-rank adaptation, focusing on addressing shortcomings of LoRA. For example, QLoRA (Dettmers et al., 2024) employs quantization technique to improve the memory consumption of LoRA. Liu et al. (2024) uses a weight decomposition of the pre-trained weight matrices to improve the performance of LoRA. Notably, Hayou et al. (2024) show that using different step sizes for different adapters in LoRA can lead to better performances. This latter work also highlight the need for algorithmic frameworks that exploit the layer-wise varying importance which our work targets to develop. However, these works are orthogonal and can be used jointly for further improvements, which we leave for future investigation. Our work is inspired by Xia et al. (2024), which uses multiple runs of LoRA to improve this algorithm.

Most related to our work is DyLoRA (Valipour et al., 2023), AdaLoRA Zhang et al. (2023) and SoRA (Ding et al., 2023). DyLoRA focuses on determining an optimal constant for setting the rank of *all* adapters in LoRA. This is in contrast to our work which aims at finding ranks for adapters that can vary across layers when given a budget on the number of trainable parameters. AdaLoRA and SoRA use different techniques to reduce (sparsify) the ranks initialized by LoRA, which in effect, gives us a rank configuration that reflects the importances of the adapted weight matrices. However, both methods violate the budget constraint when starting with a rank higher than the one initialized by LoRA. Our solution to the problem uses a sparsification method such as AdaLoRA and SoRA as a subroutine in an iterative framework, while strictly guarantees the budget constraint is satisfied.

## 3 BACKGROUND AND PROBLEM STATEMENT

### 3.1 LOW-RANK ADAPTATION

Low-rank adaptation, or LoRA, was introduced by Hu et al. (2022), as a Parameter-Efficient Fine-Tuning method, based on the assumption that the incremental update to the pretrained model has a low intrinsic rank. Specifically, given the pretrained weight $W_0 \in \mathbb{R}^{n \times d}$, LoRA considers only updates in the form of a product of two small matrices $\Delta W = BA$, where $B \in \mathbb{R}^{n \times r}$ and $A \in \mathbb{R}^{r \times d}$ with the rank $r \ll \min\{n, d\}$. We will refer to $r$ as the rank of the adapter, although $r$ is only an upper bound on the rank of $A$ or $B$. For $h = W_0 x$, the forward pass becomes

$$h = (W_0 + \Delta W)x = W_0 x + BAx.$$

In this way, the number of trainable parameters is significantly reduced from $nd$ to $r(n+d)$. For example, when the model is trained with commonly used adaptive optimizers such as Adam (Kingma, 2014), the memory required to store the optimizer's states, *e.g.*, momentums, is proportional to the number of trainable parameters. By reducing the number of trainable parameters, LoRA saves on the memory required to fine-tune the model.

In this work, we study a more generalized variation of LoRA by introducing a diagonal matrix to the update process. Let $g \in \mathbb{R}^r$ be an $r$-dimensional vector and $\mathrm{diag}(g) \in \mathbb{R}^{r \times r}$ be a diagonal matrix formed by $g$. We consider an update of the form $\Delta W = B \, \mathrm{diag}(g) \, A$. Remark here that when $g$ is set to $\mathbb{1}$, this variant becomes numerically equivalent to the standard LoRA. When $g$ is made trainable, the number of trainable parameters increases marginally by $r$, which is negligible compared to the total number of trainable variables in LoRA, *i.e.*, $r(n+d)$. This generic adjustment offers greater flexibility during training and can often lead to a more favorable optimization landscape, potentially resulting in improved performance.

### 3.2 PROBLEM STATEMENT

Suppose that we aim to use $K$ LoRA adapters for $K$ pretrained weight matrices, with corresponding ranks $r_1, \ldots, r_K$. Our research question is as follows:

> *Given a strict budget constraint on the memory required during training,*
> *how can we determine the optimal assignment of ranks for these adapters?*

Formally, let us denote the rank configuration $\mathcal{R} := (r_1, r_2, \cdots, r_K)$, and the parameters to be fine-tuned carried out by $\mathcal{R}$ as $\theta_{\mathcal{R}}$. Let $L_{W_0}(\theta_{\mathcal{R}}; \mathcal{D})$ be the loss function we optimize during the training with regard to parameters $\theta_{\mathcal{R}}$, given the pretrained weight $W_0$, and training data $\mathcal{D}$. We let $\theta_{\mathcal{R}}^*$ be a minimizer to $L_{W_0}(\theta_{\mathcal{R}}; \mathcal{D})$, *i.e.*,

$$\theta_{\mathcal{R}}^* \in \arg\min_{\theta_{\mathcal{R}}} L_{W_0}(\theta_{\mathcal{R}}; \mathcal{D}). \tag{1}$$

We measure the performance of the rank configuration $\mathcal{R} = (r_1, r_2, \cdots, r_K)$ by evaluating the objective loss values of $\theta_{\mathcal{R}}^*$ measured over the dataset $\mathcal{D}$ as follows:

$$f(\mathcal{R}) := L_{W_0}(\theta_{\mathcal{R}}^*; \mathcal{D}). \tag{2}$$

To express the memory constraint, for simplicity, we assume that all pretrained weight matrices used for adaptation have the same dimensions $n \times d$. This assumption typically holds in standard LoRA, where fine-tuning is applied to the key, query, and value matrices, *i.e.*, $W_k, W_q, W_v$ of the

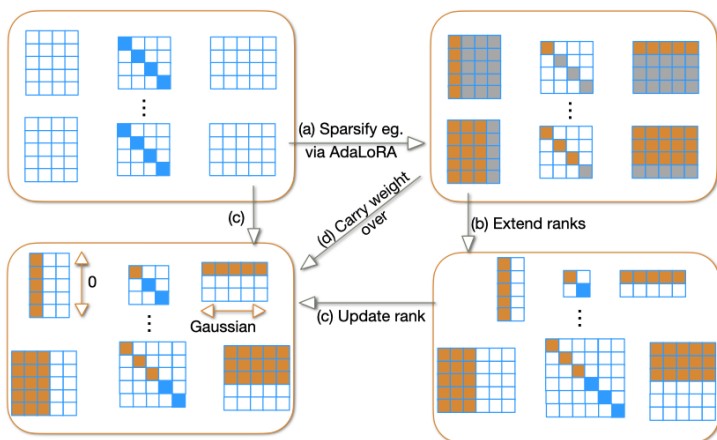

Figure 1: Visualization of an iteration in ROLoRA. Starting the iteration with an initial rank configuration (top left), the algorithm does the following: (a) Sparsify by pruning saturated adapters (grayed out in top right picture); (b) Extend the sparsified configuration; (c) Interpolate the ranks at the beginning of the iteration (top left) and the extended ranks (bottom right) to obtain the new configuration (bottom left); (d) Carry the trained weight during sparification (orange) to the new configuration and proceed to the next iteration.

self-attention modules. However, we can generalize this assumption to matrices of any size. Under this assumption, we deduce that the memory required to maintain the optimizer's states depends linearly on $\sum_{i=1}^{K} r_i$. Therefore, we express the memory constraint as:

$$\sum_{i=1}^{K} r_i \leq K \cdot R, \tag{3}$$

where $R$ is the average rank we can afford with the memory budget for each adaptor. We obtain the following main optimization problem to seek the optimal adapter configuration under strict budget:

$$\underset{\mathcal{R}}{\text{minimize}}\, f(\mathcal{R}), \text{ subject to } \sum_{i=1}^{K} r_i \leq K \cdot R. \tag{4}$$

**Remark 1**. Standard LoRA typically uses a small constant (*e.g.*, 4, 8, 16) as the value of $R$ and applies it uniformly across all adapters, *i.e.*, $r_1 = \cdots = r_K = R$. However, it neglects the varying importance of different weight matrices and has been shown to result in inferior performance compared to methods like AdaLoRA (Zhang et al., 2023), which adaptively vary the rank assignments.

**Remark 2**. To determine the optimal rank configuration, existing methods like AdaLoRA (Zhang et al., 2023) iteratively prune redundant adapter variables until convergence. However, these methods require starting with larger adapter ranks, which directly violates the memory constraint outlined in (3) during training. In contrast, our method adheres to the memory constraint from the outset and effectively discovers the optimal rank configuration, provides significant practical advantages under limited resource situations and distinguishes it from existing approaches.

Efficiently solving problem (4) is challenging for several reasons. First, accurately evaluating the performance measure $f(\mathcal{R})$ is computationally expensive. Second, standard optimization tools, such as gradient methods, struggle with discrete variables, while naive search is prohibitively expensive due to the size of the search space. In the next section, we will introduce a simple but effective method that is delicately designed to solve problem (4) efficiently.

## 4 ALGORITHM

---

**Algorithm 1** ROLoRA: Rank Optimized Low-Rank Adaptation

---

1: **Input**: Target average rank $R$
2: Initialize $r_1 = \ldots r_K = R, \gamma = 0$
3: Initialize the adapters, initialize $\mu$ the parameter of SPARSIFY algorithm according to SCHEDULE algorithm
4: **for** iteration $t = 1 \ldots T$ **do**
5:     Let $(s_1, \ldots, s_K), \theta \leftarrow$ SPARSIFY$((r_1, \ldots, r_K); \theta; \mu)$ (*e.g.* via AdaLoRA)
6:     Let $S = \frac{1}{K}(s_1 + \cdots + s_K); (s_1, \ldots, s_K) \leftarrow$ round $\left((s_1, \ldots, s_K) \times \frac{R}{S}\right)$
7:     Update $r_i = (1 - \gamma)r_i + \gamma s_i$, for all $i \in [K]$ for $\gamma = 1/t$
8:     Update current model by EXTEND$((r_1, \ldots, r_K); \theta)$
9:     Update $\mu \leftarrow$ SCHEDULE$(\mu; t)$
10: **end for**
11: **return** $(r_1, \ldots, r_K)$

---

In this section, we describe our algorithm ROLoRA to solve Problem (4) shown in Algorithm 1. ROLoRA is an iterative method to automatically discover an effective rank configuration. After proper initialization, ROLoRA begins with a given adapter rank configuration, then iteratively identifies saturated and underfitting weights. It adjusts the adapter rank assignment by performing pruning and growth operations, and is delicately structured within the Frank-Wolfe algorithmic framework to leverage potential theoretical advantages. Each main step is elaborated as follows, which is also demonstrated in Figure 1.

**Sparsifying / pruning the ranks** (Step (a) in Figure 1). Given a rank configuration $\mathcal{R} = (r_1, \ldots, r_K)$, certain weight matrices may become saturated during further fine-tuning, meaning they might not require as many trainable parameters. To address this, we introduce a sparsification or pruning operator to identify redundant ranks across the adapter sets. In Algorithm 1, we flexibly employ any rank sparsification algorithm for the SPARSIFY operator, such as AdaLoRA (Zhang et al., 2023) and SoRA (Ding et al., 2023) or pruning approaches based on saliency scores (Chen et al., 2023b; 2024). Without loss of generality, we assume that SPARSIFY takes in a pruning aggressiveness hyper-parameter $\mu$ to produce a new trial rank configuration $\mathcal{S} = (s_1, \cdots, s_K)$. $\mathcal{S}$ reflects more accurately than $\mathcal{R}$ the (relative) importance of the corresponding weight matrices (Line 5).

**Reassign and grow the ranks** (Step (b) in Figure 1). In addition to certain adapters requiring fewer ranks due to saturation, some adapters may need their capacity increased instead to learn more effectively. Therefore, it is essential to identify which adapters should grow to meet this demand. To achieve this, we use the signals from the pruned rank configuration. Typically, the ranks that remain unchanged in $\mathcal{S}$ indicate that these adapters require more parameters, as no redundancy is detected. However, because we operate under a strict memory budget, the amount of growth must strictly adhere to this constraint. Taking these factors into account, we scale up $\mathcal{S}$ to align with the target average rank $R$ (Line 6).

*Intuitive connection to gradient estimation to $f$.* The sparsification and growth steps in our algorithm effectively identify adapters that require more capacity or exhibit redundancy. Consequently, it builds on the intuition that the new rank configuration output $\mathcal{S}$ suggests a favorable search direction that can enhance the performance measure $f(\mathcal{R})$ of the current rank configuration $\mathcal{R}$.

**Update the ranks and adapters** (Step (c-d) in Figure 1). Once obtaining the new trial configuration $\mathcal{S}$, we update the current configuration $\mathcal{R}$ by moving it towards $\mathcal{S}$, *i.e.*, taking an interpolation step with step size $\gamma$ (Step (c)). This step is a reminiscence of the update step in the Frank-Wolfe algorithm (Frank et al., 1956) for solving constrained optimization problems. To update the adapter weights, we can carry over the trained adapters obtained from the SPARSIFY algorithm (Step (d)). Since the some of trial ranks in $\mathcal{S}$ are increased during the rank growth step, additional rows and columns will be added to the up and down projection matrices, as well as additional entries to the diagonal matrix, requiring only the initialization of these extended parts. These extended parts are initialized in the same way as LoRA adapters so that their product is 0 to ensure computational invariance after the adapter growth.

**Scheduling the sparsification**. After each sparsification-growth-update step, we update the sparsification parameter of the SPARSIFY operator via SCHEDULE. One approach is to keep the sparsifi-

cation parameter as a constant. Alternatively, we can employ a sparsification schedule that gradually reduces the pruning aggressiveness over time. Intuitively, as the model converges to a better rank configuration over time, it requires fewer adjustments to determine an update direction.

## 5 OUTLINE FOR CONVERGENCE ANALYSIS

In this section, we present the outline for the convergence analysis of ROLoRA to solve the target problem (4). Towards this end, we first show that under certain assumptions on the SPARSIFY operator, Algorithm 1 will improve the performance measure $f$ over iterations.

**Assumption 1.** *The* SPARSIFY$(\mathcal{R}; \theta; \mu)$ *operator has the following properties: 1. During the course of the algorithm,* SPARSIFY *satisfies the memory budget, indicated by $\sum_i r_i$. 2. For any rank configuration $\mathcal{R}$ and initial weight $\theta$, there exists a sparsification parameter $\mu$ such that if $\mathcal{S}, \theta \leftarrow$* SPARSIFY$(\mathcal{R}; \theta; \mu)$ *then $f(\mathcal{S}) \leq f(\mathcal{R})$.*

We interpret this assumption as follows. As a rank configuration can contain redundant information, the SPARSIFY operator with a well chosen sparsification parameter can remove this redundancy without sacrificing the model performance. In the trivial case, we can choose $\mu$ so that no sparsification happens, thus the loss function is not increased (yet in which case the algorithm will terminate without updating the rank configuration). In practice, if the model has saturated weights, keeping fine-tuning these weights can easily result in overfitting problem, which further degrades the model performance. Hence, yielding proper sparsity over the adapters can promote the model performance instead, *i.e.*, $f(\mathcal{S}) \leq f(\mathcal{R})$.

We then show the following Proposition.

**Proposition 1.** *Let $\mathcal{R}^{(0)} = (R, \ldots, R)$, and $\mathcal{R}^{(t)}$ be the rank configuration output by Algorithm 1 after iteration $t$. For all $t \geq 1$, $f(\mathcal{R}^{(t)}) \leq f(\mathcal{R}^{(0)})$.*

*Proof.* Let $\mathcal{S}^{(t)}$ be the sparsified ranks from $\mathcal{R}^{(t)}$ and $\theta^*_{\mathcal{S}^{(t)}}$ be a collection of optimal adapters weights for $\mathcal{S}^{(t)}$. We apply the EXTEND operator and initialization so that for each triple of adapters $(B^{(t)}, g^{(t)}, A^{(t)}) \in \theta^*_{\mathcal{S}^{(t)}}$ we have $B^{(t+1)} \text{diag}(g^{(t+1)}) A^{(t+1)} = B^{(t)} \text{diag}(g^{(t)}) A^{(t)}$. Therefore, we have $f(\mathcal{R}^{(t+1)}) \leq f(\mathcal{S}^{(t)}) \leq f(\mathcal{R}^{(t)})$, where the last inequality is by the assumption. $\square$

**Connection to Frank-Wolfe algorithm and Convergence rate**. We now show the connection between Algorithm 1 and Frank-Wolfe algorithm Frank et al. (1956). To minimize a function $f$ over a convex domain $\mathcal{X}$, in iteration $t$, Frank-Wolfe algorithm finds a solution in $s \in \mathcal{X}$ that minimizes $\langle s, \nabla f(x^{(t)}) \rangle$, with $x^{(t)}$ being the current solution, and update $x^{(t+1)} \leftarrow (1-\gamma)x^{(t)} + \gamma s$ with some step size $\gamma$. Jaggi (2013) shows that Frank-Wolfe with appropriate step sizes converges with rate $O(\frac{1}{T})$ for convex smooth functions; later Lacoste-Julien (2016) shows a $O(\frac{1}{\sqrt{T}})$ convergence rate of the duality gap for nonconvex smooth functions. Our algorithm is driven by this Frank-Wolfe type of update. Our problem (4) while being a discrete problem can be relaxed to a continuous convex domain[1]. Although we do not have access to the exact gradient $\nabla f(\mathcal{R})$, we approximate the update direction $s$ using the dedicated designed joint sparsification and growth step as the proxy described in Section 4. Consequently, we could incorporate with the existing theoretical framework of Lacoste-Julien (2016) to indicate that within a finite number of iterations, the proposed ROLoRA could find a high-performing rank configuration, which is further numerically demonstrated in Section 6.

## 6 EXPERIMENTS

We implement ROLoRA (Algorithm 1) for fine-tuning RoBERTa-base (Liu et al., 2019) and DeBERTa-v3-base (He et al., 2021) on the GLUE Benchmark (Wang et al., 2019) and SQUAD datasets (SQUADv1 (Rajpurkar et al., 2016) and SQUADv2 Rajpurkar et al. (2018)). We describe the Algorithm implementation and baselines below.

---

[1]For any two feasible $\mathcal{R}^1$ and $\mathcal{R}^2$ satisfying the budget constraint, their convex combination still satisfies the budget constraint since $\sum_{i=1}^K \left(\lambda r_i^1 + (1 - \lambda)r_i^2\right) \leq \lambda K \cdot R + (1 - \lambda)K \cdot R = K \cdot R$ for any $\lambda \in [0, 1]$.

Table 1: Results for RoBERTa-base and GLUE benchmark. The target average rank is $R = 8$. For AdaLoRA, the initial average rank is set to 12. † represents the vanilla version of the algorithm. ⋆ means the algorithm is executed in $T = 3$ iterations; in each iteration, the model is initialized to the best checkpoint. We report Pearson correlation for STSB, Matthews correlation for CoLA and Accuracy for the remaining datasets. We report the mean and standard deviation across 5 runs.

| Method | MRPC | RTE | CoLA | QNLI | MNLI | QQP | SST2 | STSB |
|--------|------|-----|------|------|------|-----|------|------|
| LoRA† | $89.90_{\pm.73}$ | $78.70_{\pm1.21}$ | $64.34_{\pm.78}$ | $93.08_{\pm.20}$ | $87.57_{\pm.11}$ | $90.92_{\pm.10}$ | $95.34_{\pm.27}$ | $90.79_{\pm.16}$ |
| LoRA⋆ | $89.36_{\pm.80}$ | $79.21_{\pm1.30}$ | $64.91_{\pm.95}$ | $93.22_{\pm.18}$ | $87.60_{\pm.09}$ | $91.21_{\pm.09}$ | $\mathbf{95.62}_{\pm.21}$ | $90.96_{\pm.11}$ |
| AdaLoRA† | $88.53_{\pm.90}$ | $76.10_{\pm.42}$ | $58.87_{\pm.46}$ | $93.56_{\pm.07}$ | $87.57_{\pm.06}$ | $90.03_{\pm.04}$ | $94.84_{\pm.07}$ | $90.26_{\pm.12}$ |
| AdaLoRA⋆ | $89.01_{\pm.36}$ | $79.42_{\pm.65}$ | $62.62_{\pm.81}$ | $\mathbf{93.72}_{\pm.02}$ | $87.66_{\pm.06}$ | $90.42_{\pm.05}$ | $95.18_{\pm.21}$ | $90.92_{\pm.09}$ |
| ROLoRA | $\mathbf{90.15}_{\pm.33}$ | $\mathbf{81.73}_{\pm.37}$ | $\mathbf{64.98}_{\pm1.20}$ | $93.71_{\pm.07}$ | $\mathbf{87.76}_{\pm.08}$ | $\mathbf{91.25}_{\pm.04}$ | $95.30_{\pm.15}$ | $\mathbf{91.18}_{\pm.13}$ |

Table 2: Results for DeBERTa-base and GLUE benchmark. The target average rank is $R = 8$. For AdaLoRA, the initial average rank is set to 12.

| Method | MRPC | RTE | CoLA | QNLI | MNLI | QQP | SST2 | STSB |
|--------|------|-----|------|------|------|-----|------|------|
| LoRA† | $90.74_{\pm.39}$ | $86.43_{\pm.98}$ | $71.16_{\pm1.49}$ | $94.24_{\pm.27}$ | $90.32_{\pm.06}$ | $92.23_{\pm.04}$ | $96.54_{\pm.09}$ | $90.75_{\pm.16}$ |
| LoRA⋆ | $90.88_{\pm.42}$ | $87.65_{\pm1.03}$ | $70.95_{\pm.45}$ | $94.40_{\pm.14}$ | $90.39_{\pm.04}$ | $92.34_{\pm.06}$ | $96.51_{\pm.21}$ | $90.79_{\pm.19}$ |
| AdaLoRA† | $91.08_{\pm.20}$ | $87.22_{\pm.37}$ | $70.39_{\pm.59}$ | $94.70_{\pm.11}$ | $90.76_{\pm.04}$ | $91.89_{\pm.02}$ | $96.24_{\pm.17}$ | $91.17_{\pm.15}$ |
| AdaLoRA⋆ | $91.27_{\pm.12}$ | $87.80_{\pm.27}$ | $\mathbf{71.78}_{\pm.73}$ | $\mathbf{94.72}_{\pm.03}$ | $\mathbf{90.93}_{\pm.05}$ | $92.17_{\pm.03}$ | $96.61_{\pm.16}$ | $91.44_{\pm.13}$ |
| ROLoRA | $\mathbf{91.72}_{\pm.65}$ | $\mathbf{88.66}_{\pm1.30}$ | $71.17_{\pm.61}$ | $\mathbf{94.72}_{\pm.07}$ | $90.78_{\pm.05}$ | $\mathbf{92.38}_{\pm.05}$ | $\mathbf{96.70}_{\pm.22}$ | $\mathbf{91.48}_{\pm.18}$ |

**Models.** We use pre-trained RoBERTa-base (Liu et al., 2019) and DeBERTa-v3-base (He et al., 2021) publicly available on HuggingFace Transformers library (Wolf, 2019). For the former, we fine-tune query and value matrices. For the latter, we fine-tune key, query and value matrices.

**Sparsification Algorithm and Scheduling**. We use AdaLoRA Zhang et al. (2023) as the SPARSIFY operator. We start with the level of sparsify that reduces the average rank to $R/2$ and gradually increase the average rank after sparsification by a factor of $\left(1 + \frac{1}{T+t}\right)$ after iteration $t$.

**Number of iterations**. We use a small number of iterations to balance the tradeoff between the improvement in the model performance and the increase in the training time. In all experiments, we use three iterations ($T = 3$).

**Baselines**. We compare our algorithm with two algorithms LoRA (Hu et al., 2022) and AdaLoRA(Zhang et al., 2023). For LoRA, we use the same target average rank $R$ across all adapters. For AdaLoRA, we initialize the rank configuration the same across all adapters to be $1.5R$ as recommended in Zhang et al. (2023) and sparsify it to the target average rank $R$. However, one should also note that by initializing the total ranks to $1.5RK$, this baselines violates the memory constraint. We also compare with two other baselines by repeating LoRA and AdaLoRA $T$ iterations. In each iteration, the model is initialized to the best checkpoint so far.

## 6.1 GENERAL LANGUAGE UNDERSTANDING EVALUATION (GLUE) BENCHMARK

**Datasets**. The GLUE Benchmark is a collection of natural language understanding tasks, consisting of two single-sentence classification tasks, three similarity and paraphrase tasks and four natural language inference tasks.

**Implementation details**. In our experiment, the target average rank is set to $R = 8$. The initial average rank for AdaLoRA is set to 12. We reuse hyper-parameters such as learning rate, sparsification parameters, etc, as recommended in the original papers by Hu et al. (2022); Zhang et al. (2023).

**Main results**. We report the performances of our algorithm in comparison with the baselines using RoBERTa-base (Liu et al., 2019) in Table 1 and DeBERTa-v3-base (He et al., 2021) in Table 2. For both models, almost across the board, ROLoRA outperforms both vanilla LoRA and AdaLoRA

Table 3: Results for RoBERTA-base on MRPC, RTE and CoLA when the target average rank $R = 16$ and $R = 4$. In the former case, the initial rank of AdaLoRA is set to 24, and in the latter 6.

| Method | $R = 16$ | | | $R = 4$ | | |
|---|---|---|---|---|---|---|
| | MRPC | RTE | CoLA | MRPC | RTE | CoLA |
| LoRA$^\dagger$ | $88.77_{\pm.57}$ | $77.77_{\pm1.34}$ | $63.21_{\pm.94}$ | $89.31_{\pm.59}$ | $79.23_{\pm1.22}$ | $64.74_{\pm.53}$ |
| LoRA$^\star$ | $89.16_{\pm.73}$ | $77.83_{\pm1.24}$ | $63.95_{\pm.86}$ | $89.85_{\pm.65}$ | $80.43_{\pm.77}$ | $\mathbf{65.99}_{\pm.99}$ |
| AdaLoRA$^\dagger$ | $87.01_{\pm.31}$ | $75.67_{\pm.67}$ | $58.50_{\pm.71}$ | $88.33_{\pm.74}$ | $77.55_{\pm2.11}$ | $59.59_{\pm.44}$ |
| AdaLoRA$^\star$ | $89.71_{\pm.35}$ | $79.49_{\pm1.24}$ | $61.51_{\pm.62}$ | $89.26_{\pm.39}$ | $80.43_{\pm.98}$ | $63.07_{\pm.30}$ |
| ROLoRA | $\mathbf{90.49}_{\pm.72}$ | $\mathbf{80.36}_{\pm1.32}$ | $\mathbf{63.98}_{\pm.83}$ | $\mathbf{90.00}_{\pm.33}$ | $\mathbf{82.38}_{\pm.48}$ | $65.57_{\pm1.26}$ |

Table 4: Average rank of the output adapters for Experiment shown in Table 1, using RoBERTA-base, for target rank $R = 8$. For LoRA and AdaLoRA, the average output rank is fixed to $R$. For ROLoRA, we report the mean across 5 runs.

| Method | MRPC | RTE | CoLA | QNLI | MNLI | QQP | SST2 | STSB |
|---|---|---|---|---|---|---|---|---|
| LoRA & AdaLoRA | | | | 8.0 | | | | |
| ROLoRA | 5.5 | 5.9 | 7.3 | 4.9 | 8.0 | 6.4 | 6.6 | 6.5 |

(marked with † in Tables 1 and 2). On RTE dataset, for example, for both models, ROLoRA shows more than 1.5% improvement compared with vanilla LoRA and AdaLoRA. This shows that setting identical ranks for all layers as in the standard LoRA is clearly a sub-optimal strategy. Even when we are allowed to violate the budget constraint and go beyond the search space as does AdaLoRA, a single step sparsification algorithm does not result in much improvements. We should also note that, while increasing the number of iterations for LoRA and AdaLoRA (marked with ⋆ in Tables 1 and 2) leads to some improvements compared with the vanilla versions, for most instances, ROLoRA still edges out on the model performance.

**Varying target average rank**. In Table 3, we report the performances of our algorithm and the baselines using RoBERTa-base when we use the target average rank $R = 4$ and $R = 16$. We discover the same pattern as when $R = 8$, showing that the ranks discovered by ROLoRA can significantly improve LoRA.

**Analysis of the output rank**. In Table 4, we report the average ranks of adapters output by each algorithm. Note that for LoRA and AdaLoRA, the average output rank is fixed to $R$, which means there is no saving in the memory. On the other hand, ROLoRA can output adapters with significantly lower ranks. For example, for the QNLI dataset, the average rank of the output adapters is reduced by 62% while ROLoRA still achieve 0.6% improvement in the performance compared with LoRA. This rank reduction can both benefit memory storage and computational time at inference. This benefit comes inherently from the way the algorithm works by always maintaining the number of trainable parameters within the budget and continually sparsifying and updating the rank configuration.

**Varying sparsification algorithm**. We experiment with SoRA Ding et al. (2023) as the SPARSIFY algorithm and show the algorithm performance in Table 5. This version of ROLoRA generally demonstrates improvement over LoRA. However, we should note that since SoRA uses a thresholding based strategy for pruning the ranks as opposed to a ranking based strategy used by AdaLoRA, it is more difficult to control the sparsity of the solution. For this reason, in this experiment, we keep the same threshold across all iterations of the algorithm (for the SCHEDULE operator). The performance of the algorithm could be improved with a better SCHEDULE operator.

## 6.2 QUESTION-ANSWERING (SQUAD) BENCHMARK

**Datasets**. SQUAD consists of two datasets for a Question-Answering task, in which we predict the probability of each token being the start and end of the answer to the question.

Table 5: Results when varying the sparsification algorithm, using RoBERTA-base model. We compared our algorithms with the vanilla (marked with †) and iterative (marked with ⋆) versions of LoRA (part of Table 1). Superscript $^A$ represents ROLoRA with AdaLoRA as the SPARSIFY operator, and $^B$ represents ROLoRA with SoRA as the SPARSIFY operator.

| Method | MRPC | RTE | CoLA | SST2 |
|---|---|---|---|---|
| LoRA$^†$ | $89.90_{\pm.73}$ | $78.70_{\pm1.21}$ | $64.34_{\pm.78}$ | $95.34_{\pm.27}$ |
| LoRA$^⋆$ | $89.36_{\pm.80}$ | $79.21_{\pm1.30}$ | $64.91_{\pm.95}$ | $\mathbf{95.62}_{\pm.21}$ |
| ROLoRA$^A$ | $\mathbf{90.15}_{\pm.33}$ | $\mathbf{81.73}_{\pm.37}$ | $64.98_{\pm1.20}$ | $95.30_{\pm.15}$ |
| ROLoRA$^S$ | $89.90_{\pm.57}$ | $80.0_{\pm1.52}$ | $\mathbf{65.14}_{\pm1.33}$ | $95.34_{\pm.22}$ |

Table 6: Results for DeBERTa-v3-base on SQUADv1 and SQUADv2 datasets when the target average rank is $R = 8$ and $R = 4$. We report Exact Match for both datasets.

| Rank | $R = 8$ | | $R = 4$ | |
|---|---|---|---|---|
| Method | SQUADv1 | SQUADv2 | SQUADv1 | SQUADv2 |
| LoRA$^†$ | 87.73 | 84.71 | 87.71 | 85.05 |
| LoRA$^⋆$ | 87.85 | 84.61 | 87.73 | 85.13 |
| AdaLoRA$^†$ | 87.90 | 85.72 | 87.74 | 85.26 |
| AdaLoRA$^⋆$ | 88.25 | **86.20** | 87.90 | 85.35 |
| ROLoRA | **88.29** | 85.87 | **87.97** | **85.60** |

**Implementation details**. We experiment with the target average rank set to $R = 8$ and $R = 4$. In the former case, the initial average rank for AdaLoRA is set to 12, and in the latter, it is set to 6.

**Main results**. We report the results using DeBERTa-base model. In both cases $R = 8$ and $R = 4$, our algorithm outperforms both vanilla LoRA and AdaLoRA. The improvement over LoRA is more than $1.1\%$ on SQUADv2, when $R = 8$. Even though in this case, AdaLoRA with $T$ iterations shows a better performance, we should note that AdaLoRA does not adhere to the memory constraint.

## 6.3 ABLATION STUDY

We conduct an in-depth study to show why ROLoRA outperforms LoRA and AdaLoRA.

To understand how ROLoRA assigns ranks, we plot the rank distribution of the output adapters for DeBERTa-v3-base on CoLA and QNLI datasets in Figure 2(a) (same experimental results in Table 2). We compare these distributions with the configurations discovered by AdaLoRA in Figure 2(b), which also serves as the base SPARSIFY operator in our algorithm. We observe the following:

(1)    ROLoRA finds a configuration that allows ranks to go beyond the upper limit set by AdaLoRA. In particular, in the plot for CoLA in Figure 2(a), some adapters have ranks higher than the initial rank 12 set by AdaLoRA in the same experiment in Figure 2(b). This means, ROLoRA can go beyond the search space set by a vanilla sparsification algorithm, and thereby achieves improvements.

(2)    ROLoRA focuses significantly on the value matrices. The overall ranks assigned to the value matrices are significantly higher than those to the key and query matrices. Compared with AdaLoRA, which shows a similar pattern, ROLoRA further accentuates this difference.

We further investigate this latter point to see the role each type of matrices plays in fine-tuning.

In Figure 3 we plot the gradient norm of the pretrained DeBERTA-v3-base model by making one backward pass over the corresponding datasets (CoLA and QNLI). We can see that in each layer, the value matrices have much larger gradient norms than the others. Intuitively, this implies that these matrices are much more sensitive and thus require higher level of tuning. Therefore, one could expect that the value matrices play a more important role in fine-tuning and by assigning higher ranks to the adapters for them relatively to the other matrices, we could achieve a better performance.

To verify this hypothesis, we train separately three sets of adapters: for each type of matrices (key, value, query), we inject a LoRA adapter for each matrix of that type of rank $R = 8$ and fine-tune the model. We report their performances in Table 7. The result in Table 7 shows that fine-tuning only value matrices can give us better performances than doing so with key and query matrices. This confirms out hypothesis that value matrices play a more important role in fine-tuning LLMs.

Returning to Figure 2, the performance of ROLoRA can by explained by its accentuation on the ranks of value matrices.

Table 7: Results when fine-tuning key, query and value matrices separately with LoRA adapters and rank $R = 8$.

| | Value | Key | Query |
|---|---|---|---|
| CoLA | **69.7** | 68.9 | 67.1 |
| QNLI | **94.3** | 93.7 | 93.7 |

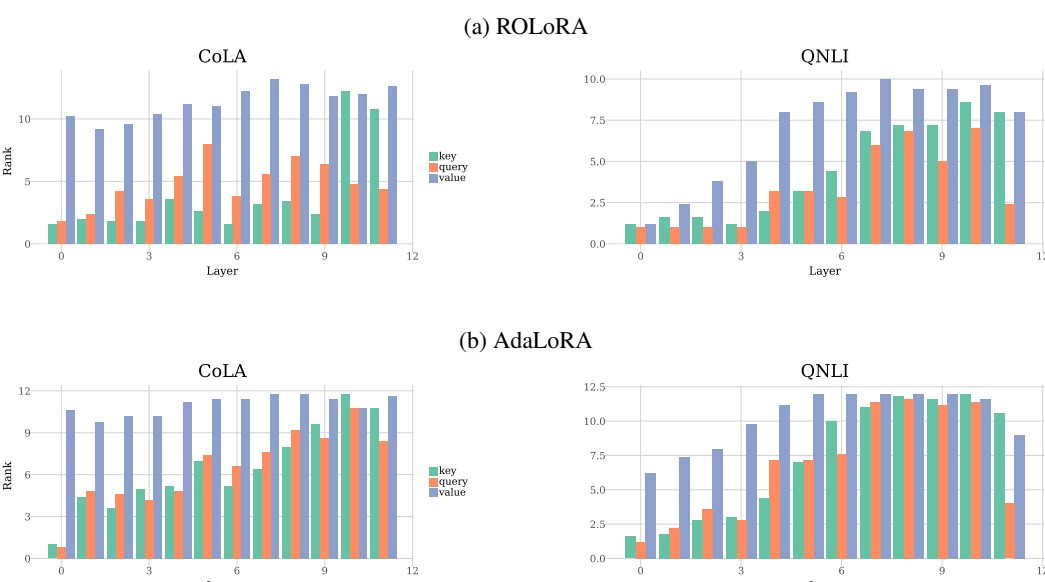

Figure 2: Distribution of ranks across layers obtained by (a) ROLoRA and (b) AdaLoRA on CoLA and QNLI using DeBERTA-v3-base. Both algorithms give higher rank to adapters for value matrices, though ROLoRA further accentuates this feature.

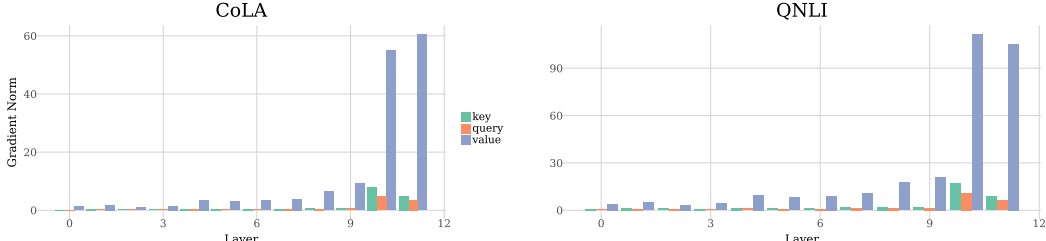

Figure 3: Distribution of gradient norms across of the pre-trained model across layers. Value matrices have higher significantly gradient norms than key and query matrices, implying that these matrices are much more sensitive and require more tuning.

## 7 FUTURE WORK AND CONCLUSION

In this work, we propose a novel iterative algorithm that optimizes rank configurations to enhance the performance of the popular fine-tuning algorithm LoRA. Our approach leverages a new insight that value matrices in the transformer architecture are more critical for fine-tuning on downstream tasks than key and value matrices. We leave further investigation on the applications of the our method and its theoretical convergence guarantee to future work.

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

Table 8: Hyperparameters are identical for all methods.

| Model | Hyperparams | MRPC | CoLA | RTE | QNLI | MNLI | SST2 | QQP | STSB |
|---|---|---|---|---|---|---|---|---|---|
| RoBERTa-base | # Epochs | 30 | 25 | 50 | 25 | 25 | 30 | 25 | 25 |
| | Warm-up Ratio | | | | 0.06 | | | | |
| | Batch-size | 32 | 32 | 32 | 16 | 16 | 16 | 16 | 32 |
| | $\alpha$ | | | | 8 | | | | |
| | Learning rate | 1e-3 | 5e-4 | 5e-4 | 5e-4 | 5e-4 | 5e-4 | 5e-4 | 5e-4 |
| | Max Seq. Length | | | | 128 | | | | |
| DeBERTa-v3-base | # Epochs | 30 | 30 | 50 | 25 | 30 | 25 | 25 | 25 |
| | Warm-up Ratio | | | | 0.06 | | | | |
| | Batch-size | 32 | 32 | 32 | 16 | 16 | 16 | 32 | 32 |
| | $\alpha$ | | | | 8 | | | | |
| | Learning rate | | | | 5e-4 | | | | |
| | Max Seq. Length | | | | 128 | | | | |

Table 9: Sparsification parameters for AdaLoRA and ROLoRA.

| Model | Hyperparams | MRPC | CoLA | RTE | QNLI | MNLI | SST2 | QQP | STSB |
|---|---|---|---|---|---|---|---|---|---|
| RoBERTa-base | $\gamma$ | 0.1 | 0.5 | 0.3 | 0.1 | 0.1 | 0.1 | 0.1 | 0.1 |
| | $t_i$ | 600 | 600 | 600 | 2000 | 8000 | 600 | 8000 | 600 |
| | $t_f$ | 1800 | 3500 | 1800 | 8000 | 50000 | 22000 | 25000 | 2000 |
| | $\Delta_T$ | 1 | 10 | 1 | 100 | 100 | 100 | 100 | 10 |
| DeBERTa-v3-base | $\gamma$ | 0.1 | 0.5 | 0.3 | 0.1 | 0.1 | 0.1 | 0.1 | 0.1 |
| | $t_i$ | 600 | 600 | 600 | 2000 | 8000 | 600 | 8000 | 600 |
| | $t_f$ | 1800 | 3500 | 1800 | 8000 | 50000 | 22000 | 25000 | 2000 |
| | $\Delta_T$ | 1 | 10 | 1 | 100 | 100 | 100 | 100 | 10 |

## A DATASET DETAILS

**GLUE benchmark.** GLUE benchmark consists of: MNLI (inference Williams et al. (2018)), SST-2 (sentiment analysis, Socher et al. (2013)), MRPC (paraphrase detection, Dolan & Brockett (2005)), CoLA (linguistic acceptability, Warstadt et al. (2018)), QNLI (inference, Rajpurkar et al. (2018)), QQP8 (question-answering), RTE (inference), and STS-B (textual similarity, Cer et al. (2017)).

**SQUAD.** SQUAD consists of two Question-Answering datasets: SQUADv1 (Rajpurkar et al., 2016) and SQUADv2 Rajpurkar et al. (2018).

## B HYPERPARAMETERS

### B.1 GLUE BENCHMARK

Hyper-parameters choices are shown in Tables 8 and 9.

### B.2 SQUAD BENCHMARK

Hyper-parameters choices are shown in Tables 10 and 11.

Table 10: Hyperparameters are identical for all methods.

| Hyperparams | SQUADv1 SQUADv2 |
|---|---|
| # Epochs | 12 |
| Warm-up | 1 |
| Batch-size | 16 |
| $\alpha$ | 8 |
| Learning rate | 1e-3 |
| Max Seq. Length | 128 |

Table 11: Sparsification parameters for AdaLoRA and ROLoRA.

| Hyperparams | SQUADv1 SQUADv2 |
|---|---|
| $\gamma$ | 0.01 |
| $t_i$ | 5000 |
| $t_f$ | 25000 |
| $\Delta_T$ | 100 |

