# OpenReview forum: "ROLoRA: Rank Optimization  for Low-Rank Adaptation under Memory Constraints"
_ICLR.cc/2025/Conference — Submitted to ICLR 2025_

### Official Review · Reviewer_z1pc · 2024-10-31

**Soundness:** 3
**Presentation:** 3
**Contribution:** 3
**Rating:** 5
**Confidence:** 3

**Summary:**

The paper introduces a method called ROLoRA. ROLoRA is a novel method that efficiently discovers an effective rank configuration for low-rank adaptation while strictly adhering to a constrained computational budget during training. It outperforms standard LoRA on natural language processing tasks and is practical for resource-constrained scenarios.

**Strengths:**

>The writing is good and the paper is easy to follow.

>The method can efficiently discover an effective rank configuration for low-rank adaptation.


>ROLoRA outperforms standard LoRA on some benchmarks under some model configurations.

**Weaknesses:**

> The experiments are conducted on models like RoBERTa and DeBERTa. Have any experiments been conducted on modern large language models like Llama or OPT? Including these experiments will make the method much more valuable in modern settings.

> The experimental results sometimes seem to fall behind baselines and the improvements are not that significant. For e.g., LoRA⋆ in Table 1 and AdaLoRA⋆ in Table 2.


> Any comparisons in running time and convergence speed with baselines?

**Questions:**

See weanesses.

---

> ### Author Response · Authors · 2024-11-27
>
> We thank the reviewer for the valuable feedbacks to our paper. We answer the reviewer's questions below:
>
> > 1. Have any experiments been conducted on modern large language models
> like Llama or OPT
>
> We will provide additional experimental results on Lllma-7B and Commonsense benchmark in the general comments.
>
> > 2. The experimental results sometimes seem to fall behind baselines
> and the improvements are not that significant
>
> We would like to point out that in Table 1, our algorithm is better
> than LoRA\* on almost all datasets (except SST2). In Table 2, we
> would also like to make it clear that, even though we use AdaLoRA\*
> as a baseline, the algorithm requires **1.5 times more memory** and therefore
> **exceeds the memory constraints**. This means our algorithm achieves a
> comparable performance with significantly less memory usage.
>
> > 3. running time and convergence speed with baselines
>
> As in our answer to the other Reviewers, since we use 3 iterations, we expect
> the training time to increase proportionally. However, because training
> time is not the biggest concern for us (we focused on addressing the
> memory constraint), we didn't focus on optimizing the running time
> of the algorithm. We believe that the running time could be significantly
> reduced if optimized (via techniques such as early stopping).

---

### Official Review · Reviewer_Rw6n · 2024-11-02

**Soundness:** 3
**Presentation:** 3
**Contribution:** 3
**Rating:** 6
**Confidence:** 4

**Summary:**

The paper introduces ROLoRA, a method for optimizing adapter ranks in Low-Rank Adaptation (LoRA) to enable efficient fine-tuning of large language models within memory constraints. Unlike standard LoRA, which applies a fixed rank across all layers, ROLoRA iteratively adjusts ranks by pruning overfitted adapters and expanding under fitted ones, all within a specified memory budget. Experimental results demonstrate that ROLoRA outperforms existing methods like AdaLoRA on benchmarks such as GLUE and SQuAD, particularly by optimizing ranks for crucial weight matrices in transformer layers, such as the value matrices in attention mechanisms.

**Strengths:**

Overall, this is a good work. The algorithm is well-motivated and reasonably well-explained. The method is backed by adequate empirical evidence, with results that support the effectiveness of ROLoRA over existing approaches. In particular, the ablation study is a valuable addition, demonstrating that the rank assignments are not only adaptive but also focus on key components, such as the value matrices in attention layers, which are shown to benefit from higher ranks.

**Weaknesses:**

Weaknesses

1. Insufficient References in Key Claims (Lines 139-141): Certain claims in the paper are presented without adequate referencing, reducing their credibility, for instance “can often lead to a more favorable optimization landscape”.

2. Lack of Justification for Assumption 1: The authors assume that the SPARSIFY operator maintains memory constraints and can remove redundancy without sacrificing model performance. This assumption is pivotal to the algorithm, yet it lacks theoretical backing. Providing additional rationale here would reinforce the assumption’s validity.

3. Ambiguity in Proof of Proposition 1: The proof for Proposition 1, which posits that ROLoRA iteratively improves model performance, is not entirely convincing. The algorithm currently appears heuristic, without formal assurance that each iteration yields a performance improvement similar to the EM algorithm. A clearer proof structure or additional evidence supporting iterative improvement would strengthen this point.

4. Limited Explanation of Frank-Wolfe Connection and Convergence: While the authors mention a connection to the Frank-Wolfe algorithm, it needs further explanation. It is unclear how the discrete-to-continuous transition (which seems more heuristic) impacts convergence guarantees. Further elaboration on how the theoretical aspects would still hold after this discrete to continuous transition would enhance clarity.

5. Unclear Baseline Iteration Details: The paper lacks a detailed comparison of iteration counts between ROLoRA and baseline methods like LoRA* and AdaLoRA*. It is therefore uncertain whether these baselines received an equivalent level of optimization. Including these details would facilitate a more accurate assessment of relative performance.

6. Average Rank in Table 4: Table 4 indicates that ROLoRA achieves a lower average rank than sparsification-only methods like AdaLoRA, despite ROLoRA’s additional expansion steps. The reasoning behind this outcome is unclear. A more detailed explanation of how the rank pruning and expansion operations jointly lead to this effect would clarify the results.

7. Absence of Average Rank on SQuAD Datasets: The paper presents average rank results for GLUE but omits similar data for the SQuAD datasets. Providing this information would complete the evaluation, illustrating ROLoRA’s impact on question-answering tasks.

8. Scalability Testing on Larger Models: The paper’s evaluation on smaller models leaves open questions regarding scalability. Testing on larger models, such as those with 1B or 7B parameters, would confirm if ROLoRA’s efficiency extends to more substantial architectures, making the findings more broadly applicable.

**Questions:**

Please see above.

I am curious to see how ROLoRA performs using the sparsification method from AutoLoRA (https://aclanthology.org/2024.naacl-long.282.pdf), which is built upon a similar motivation as AdaLoRA. It would be interesting to explore how this approach compares or complements the existing sparsification techniques used in this work.

**Details Of Ethics Concerns:**

Not needed

---

> ### Author Response · Authors · 2024-11-27
>
> We thank the reviewer for the careful and valuable feedback. We answer
> the reviewer's comments below.
>
> > 1. Insufficient References in Key Claims (Lines 139-141):
>
> Thank you for pointing out. For example, injecting the diagonal matrix (gating unit) is followed from SoRA (Ding et al, 2023), which provides a strict generalization of LoRA allows for both flexibility in training (via sparse optimizer) and better results.
>
> > 2. Lack of Justification for Assumption 1
>
> One can think of Assumption 1 as having a Sparsification algorithm
> that can remove completely redundant features and therefore
> does not sacrifice the model performances. However, we agree with
> the reviewer that this assumption is quite restrictive. We would also
> like to point out that, the assumption is mainly for our theoretical
> purposes; and due to the difficult nature of the problem (optimization
> with respect to the rank configuration, which is a discrete domain, with a non-convex function),
> without further restrictive assumption, the analysis is very challenging.
>
> > 3. Ambiguity in Proof of Proposition 1
>
> We would like to clarify that due to Assumption 1 (the Sparsifier
> can remove redundant features without sacrificing the model performances)
> and the fact that extending the ranks maintains the same model performances,
> our iterative routine at least does not decrease the performance and
> can yield performance improvement. We would make this clarification
> in the revision of the paper.
>
> > 4. Limited Explanation of Frank-Wolfe Connection and Convergence
>
> Indeed, the reviewer is correct that we consider the relaxed version
> of the problem to a continuous domain as a heuristic. We only consider
> the connection with the Frank-Wolfe framework for this relaxed version
> of the problem. In practice, as mentioned in the algorithm, a rounding
> step is required to obtain a discrete solution. We would make this
> clearer in the next version of the paper.
>
> > 5. Unclear Baseline Iteration Details
>
> For the baselines LoRA\* and AdaLoRA\*, we keep the iteration counts
> for both baselines T=3, the same we used in our algorithm. In each
> iteration, we keep the optimization level identical across methods
> (except the adjustments due to the differences among these approaches):
> we reinitialize the model at the best checkpoint, reinitialize
> the optimizer, and use the same number of epochs. We would make this
> clarification in the revision of the paper.
>
> > 6. Average Rank in Table 4
>
> We report the average rank of the adapters at the best checkpoint.
> It could happen that ROLoRA achieves a better performance during the
> sparsification step, instead of the rank extension step, due to a
> potential regularization effect of sparsification.
>
> > 7. Absence of Average Rank on SQuAD Datasets
>
> Thank you for your suggestion. We will provide the average rank on the SQuAD Datasets in the revision of the paper.
>
> > 8. Scalability Testing on Larger Models
>
> We will provide additional experimental results on Lllma-7B and Commonsense benchmark in the general comments.
>
> > 9. how ROLoRA performs using the sparsification method from AutoLoRA
>
> We thank the reviewer for providing the reference that we missed. We find it would be very interesting to see how other sparsification methods can incorporate with ROLoRA in practice. AutoLoRA appears to be an efficient sparsification method that could potentially improve the efficiency of ROLoRA and would be a great additional results we will add to the next version of the paper.

---

### Official Review · Reviewer_gY5x · 2024-11-03

**Soundness:** 3
**Presentation:** 2
**Contribution:** 2
**Rating:** 5
**Confidence:** 3

**Summary:**

This paper proposes ROLoRA - a new PEFT (Parameter-Efficient Fine-Tuning) method that adjusts adapter ranks for different modules and layers under a constrained memory budget. Unlike LoRA, which applies the same rank for all adapters, but similar to AdaLoRA, ROLoRA suggests that different ranks for different modules are a better approach (confirmed by experiments). Compared to AdaLoRA, ROLoRA stays within the constrained budget during training but may increase training time (though by how much is unclear from the paper). ROLoRA is an iterative method involving pruning and then expanding ranks. The approach is tested on RoBERTa-base and DeBERTa-v3-base. Experiments show improvements over LoRA and competitive performance to AdaLoRA on the GLUE and SQuAD benchmarks.

**Strengths:**

Strengths

-Clear motivation and introduction

-Relevant problem

-New iterative framework that operates within a constrained budget throughout LLM finetuning

-Interesting analysis and insights on the importance of value matrices in finetuning

**Weaknesses:**

In general, I think this is a relevant problem and interesting approach. However, the biggest issue is that I would like to understand how this approach is better than AdaLoRA. Specifically, what is the increase in finetuning time introduced by ROLoRA, and how much does AdaLoRA exceed the computational budget during fine-tuning? What if we set the computational budget to be max N in AdaLoRA and exactly N in ROLoRA? How do the two algorithms compare? I would like to understand when someone would prefer to use ROLoRA, as it is a more complex algorithm (with longer fine-tuning time). I would also like to see the method’s behavior on larger models (decoder-only) and more recent tasks. Also, there is no mention of releasing code for this framework, which raises concerns about usability.

Weaknesses in points:

-Tested only on two encoder-only models and only GLUE and SQUAD benchmarks. I think that more models, possibly a larger decoder-only model, and some more recent benchmarks would be beneficial.

-The insights about value matrices are interesting, but they seem more observational than a motivating factor for ROLoRA design. In the conclusions, the value insight is highlighted as a main takeaway, but it’s only mentioned briefly in the ablation study at the end.

-The paper mentions balancing iterations and training time, but it would be helpful to see a clear analysis showing how much training time increases.

-Figure 2 could be improved to present a better side-by-side comparison (currently, it’s difficult to read).

-Table 4 shows average ranks, but a summary of parameter counts for LoRA, AdaLoRA, and ROLoRA would clarify the overall memory savings.

-The final sections of the paper are not as comprehensive as the earlier sections. The value matrix insight is introduced very quickly, and further analysis would be useful.

**Questions:**

Questions (a few questions also in the Weaknesses section)

-How does training time increase per iteration in ROLoRA? Is it 3x normal training for 3 iterations, or is it faster?

-Given ROLoRA complexity, do you plan to release the code?

-L062-L064: Could you add citations here?

-I think this paper could benefit from a visualization of rank changes. This might offer interesting insights for future work. Is it possible to generate such a plot?

-About Frank-Wolfe framework: Could you please add more explanation of why the Frank-Wolfe framework was chosen? Can you clarify the term "delicately designed" (L023) in describing the Frank-Wolfe framework? Why “potential” in “potential theoretical guarantees”? Could you explain? (L024)

-How were the hyperparameters for the experiments chosen? This is not clear from the paper and may have an impact on the results.

---

> ### Author Response · Authors · 2024-11-27
>
> We thank the reviewer for the detailed and constructive feedback.
> We will keep on improving the paper based on the reviewer's suggestions.
> We answer the reviewer's questions below.
>
> > 1. I would like to understand how this approach is better than AdaLoRA
>
> Thank you for the question. Our outperformances are two folds.
>
> **Efficiency**. The recommended way to run
> AdaLoRA (by the original paper) is, given the budget N (the average
> rank), AdaLoRA starts with a higher budget (1.5N) and reduces the
> average rank to N. This is a clear violation of the memory budget
> we set during the training time. In contrast, the use of AdaLoRA in ROLoRA is always executed under budget N and we are able to optimize the ranks effectively. Meanwhile, our approach is not limited to AdaLoRA and is flexible to integrate with more efficient sparsification schema, such as SoRA to further enhance the efficiency.
>
> **Performance**. When AdaLoRA optimizes the ranks, it only conducts pruning strategy. The pruned rank can not be recovered. It raised the concerns that if some false positive ranks are pruned, which can not be recovered under AdaLoRA schema, consequently leading to sub-optimal results. However, our ROLoRA is iteratively pruning and growing, effectively resolve the risk of false positive rank deletion and leads to higher performance.
>
> > 2. behavior on larger models (decoder-only) and more recent tasks
>
> We will provide additional experimental results on Lllma-7B and Commonsense benchmark in the general comments.
>
> > 3. releasing code for this framework, which raises concerns about usability.
>
> We have uploaded the code. Our code is built upon the PEFT framework by HuggingFace. We will also release the code for public use.
>
> > 4. The insights about value matrices are interesting, but they seem more observational than a motivating factor for ROLoRA design
>
> Indeed, the insights about the differences among the matrices come
> only in hindsight and support our motivation that the differences
> among the matrices call for an improvement to LoRA that adapts to
> these differences. These insights are not purposefully set in the
> design of ROLoRA, but interestingly, ROLoRA outputs configurations
> that are in line with these insights. This means, the improvements by ROLoRA are not just due to an increase in the training steps,
> but from its ability to detect meaningful patterns.
>
> > 5. The paper mentions balancing iterations and training time, but
> it would be helpful to see a clear analysis showing how much training
> time increases.
>
> Since we use 3 iterations, we expect the training time to increase
> proportionally. However, as in our response to Reviewer vHgk, since
> training time is not the biggest concern for us (we focused on addressing
> the memory constraint), we didn't focus on optimizing the running
> time of the algorithm. We believe that the running time could be significantly
> reduced if optimized (via techniques such as early stopping).
>
> > 6. Figure 2 could be improved
>
> Thank you for the suggestion. We will keep on improving the presentation
> of the paper.
>
> > 7. Table 4 shows average ranks, but a summary of parameter counts for LoRA, AdaLoRA, and ROLoRA would clarify the overall memory savings.
>
> The average ranks indicate the number of parameters. For example,
> for the QQP dataset, the average rank is reduced from 8 to 6.4. This
> implies a 20\% reduction in the parameter counts in the output adapters.
>
> > 8. The final sections of the paper are not as comprehensive as the
> earlier sections. The value matrix insight is introduced very quickly,
> and further analysis would be useful.
>
> Thank you for the recommendation. We will improve this section in the revision of the paper.
>
> > 9. L062-L064: Could you add citations here?
>
> Yes, previous works that explores the varying importance of the weight
> matrices include AdaLoRA (Zhang et al., 2023) and SoRA (Ding et al.
> 2023), purely on sparsification of ranks. Both requires an initialization
> higher than the allowed memory budget.
>
> > 10. I think this paper could benefit from a visualization of rank changes.
>
> We thank the reviewer for the suggestion. Due to the number of matrices we finetune, visualizing the rank changes for all matrices is difficult. However, we could provide plots for some representative matrices.
>
> > 11. Could you please add more explanation of why the Frank-Wolfe framework was chosen?
>
> Instead of using a gradient method (with respect to the rank configuration),
> which would be very difficult to compute, we realize that the sparsified
> configuration gives us a direction to improve the
> current solution. This is closely related to the Frank-Wolf framework, although
> without further assumptions, we couldn't provide a rigorous theoretical
> proof. We  leave further theoretical investigation to future
> work.
>
> > 12. How were the hyperparameters for the experiments chosen?
>
> We followed prior works to select the hyperparameters, which are provided in the appendix.

---

> > ### Comment · Reviewer_gY5x · 2024-12-02
> >
> > Thank you for including additional results. I appreciate addressing reproducibility concerns and discussing the questions. However, as the detailed discussion about the finetuning time is missing, I am unsure how much ROLoRA increases this time in practice, which I believe is an important practical aspect.
> > While the experiments show that ROLoRA uses fewer parameters in the given setting, a fair comparison would require trying to match the parameter count with AdaLoRA to evaluate their differences. A possible experiment would be to start with the same rank N for both methods across different seeds and compare their parameter efficiency and accuracy at various runs. This could provide more clarity on their relative performance. I find this research interesting, but would require further refinement of the experimental and theoretical aspects.
> > I believe my concerns about the practical usability of this method remain. Therefore, I would keep my current score.

---

> ### Author Response · Authors · 2024-12-03
>
> We thank the reviewer for the insightful feedback.
>
> Part of the reasons why we did not directly compare with AdaLoRA initialized to the same as ROLoRA is because it is unclear to what rank we should sparsify AdaLoRA. The sparsity scheduler used in ROLoRA addresses this problem. We report below the performance on the same benchmark by training AdaLoRA initialized to the same as ROLoRA (rank 16) and sparsify it to rank 10 for comparison.
>
> | |  PIQA | SIQA | ARC-e | ARC-c | WinoGrande | BoolQ | OBQA | Avg. |
> | :--- | :---: | :---: |  :---: | :---: | :---: | :---: | :---: | :---: |
> | AdaLoRA | 78.72 | 78.10 | 80.18 | 64.06 | 79.95 | 69.63 | 76.8 | 75.34 |
> | Ours | 79.32 | 78.81 | 81.10 | 65.36 | 80.42 | 71.25 | 78.4 | 76.38 |

---

### Official Review · Reviewer_vHgk · 2024-11-04

**Soundness:** 3
**Presentation:** 2
**Contribution:** 2
**Rating:** 5
**Confidence:** 3

**Summary:**

This paper proposes ROLoRA, an iterative algorithm for optimizing rank configurations in LoRA-based fine-tuning of large language models. The key insight is adaptively adjusting ranks across different weight matrices while strictly adhering to memory constraints. The method iteratively sparsifies saturated adapters and grows underfitted ones within a Frank-Wolfe optimization framework. The authors evaluate ROLoRA on GLUE and SQUAD benchmarks using RoBERTa-base and DeBERTa-v3-base models, demonstrating improved performance over standard LoRA and AdaLoRA baselines while maintaining lower memory usage. A notable finding is that value matrices in transformer architectures require higher adaptation capacity compared to key/query matrices.

**Strengths:**

- The problem formulation effectively addresses a practical limitation of LoRA - the need to determine optimal rank configurations under strict memory constraints. However, although the problem is clearly defined, I'm not sure if it is important. Given that LoRA already works well and is simple and effective, do we need to further constrain the budget to achieve trivial improvements?
- The iterative optimization approach is theoretically grounded in the Frank-Wolfe framework, making the method more principled than heuristic alternatives.
- The empirical analysis is comprehensive, with clear ablation studies that reveal insights about the varying importance of different weight matrices.

**Weaknesses:**

- If I remember correctly, I have run AdaLoRA and even AdaLoRA is very time-consuming. Thus my major concern is the computational efficiency of the method. Does RoLoRA also face issues with computational overheads? Will there be detailed analysis? Given that RoLoRA's improvements over LoRA are not particularly significant, it's difficult to assess the merits of this method without detailed computational analysis.
- I know that many LoRA works use the same experimental settings as this one, and it is convenient to make comparisons. However, I think using these models and benchmarks in 2024 might be somewhat outdated, as their behaviors may change as model capabilities continue to improve.

**Questions:**

- Figure 1 can be further improved. For example, add legends to clarify which kinds of grids represent which kinds of matrices.

---

> ### Author Response · Authors · 2024-11-27
>
> We thank the reviewer for the detailed and valuable feedback. We answer the reviewer's concern below.
>
> > 1. Given that LoRA already works well and is simple and effective, do we need to further constrain the budget to achieve trivial improvements?
>
> Thank you for the great question. Our motivations are two folds.
>
>
> At first, imposing a budget constraint comes from the important need to enable fine-tuning on small edge devices, which have a small memory capacity for both training and storing adapters. Such scenarios is becoming increasingly important with a lot of down-stream applications. For example, there is an trend of employing SLMs on-devices, where having personalization and maintaining privacy is becoming hard demands. In such situations, conducting fine-tuning under an restricted constraint is a necessity.
>
> Secondly, the quality of fine-tuning matters, since it will directly affects the user-experience of SLMs. Although people could reduce the ranks of Lora to meet the tight budget constraint, Lora is sub-optimal due to lack of consideration of knowledge distributions. In contrast, our approach could effectively optimize the rank distribution under memory constraint and deliver better performance.
>
> Based on the above, we believe that our work studies an important problem and deliver significant technical and practical contributions.
>
> > 2. AdaLoRA is very time-consuming. Thus my major concern is the computational efficiency of the method.
>
> Thank you for the question. We would like to highlight that our approach is flexible to any sparsification subroutine. AdaLoRA is one of them and selected due to its popularity. We did observe an increase in the computation overheads when using AdaLoRA. But we can integrate with other variants, such as SoRA (Ding et al, 2023), which is a more efficient sparsification method that doesn't increase computation overheads.
>
> > Will there be detailed analysis?
>
> In our experiment, we use T=3 iterations and expect the training time to increase proportionally. However, since training time is not the
> biggest concern for us (we focused on addressing the memory constraint), we didn't focus on optimizing the running time of the code. We believe that the running time could be significantly reduced if optimized (via techniques such as early stopping).
>
> > 3. using these models and benchmarks in 2024 might be somewhat outdated
>
> We will provide additional experimental results on Lllma-7B and Commonsense benchmark in the general comments.
>
> **Questions**
>
> > 1. Figure 1 can be further improved.
>
> We thank the reviewer for pointing out the missing details in Figure 1. We will improve it as the reviewer suggested.

---

### Meta-Review · Area_Chair_sbC9 · 2024-12-15

**Metareview:**

In this paper, the authors proposed a new low-rank adaptation method for LLMs, which efficiently discovers ranks for different layers.

Though some promising results are presented, there are several major concerns raised by the reviewers. 1, The effectiveness of the proposed method on "larger" models is unknown, which makes the results less convincing. 2, The time complexity of the proposed method is not fully analyzed. 3. How the proposed method is better than existing methods is not very clear.

This is borderline paper. As some major concerns remain after the rebuttal, a moderate revision is required. Thus, based on the current shape, this paper is not ready for publication.

**Additional Comments On Reviewer Discussion:**

The reviewers' concerns about the time complexity and effectiveness of the proposed method on LLMs remain.

---

### Decision · Program_Chairs · 2025-01-22

Reject